# Very high-speed running (VHSR) profile in elite female football: An update

Aratz Olaizola[1]*, Ibai Errekagorri[1], Karmele Lopez-de-Ipina[2,3,4], Pilar María Calvo[3,4], Julen Castellano[1,5]

1 Department of Physical Education and Sport, Faculty of Education and Sport, University of the Basque Country (UPV/EHU), Vitoria-Gasteiz, Spain, 2 Department of Psychiatry, Cambridge Neuroscience, University of Cambridge, Cambridge, United Kingdom, 3 EleKin Research Group, Department of System Engineering and Automation, University of the Basque Country (UPV/EHU), Donostia-San Sebastian, Spain, 4 Department of Architecture and Technology of Computers, Computer Science Faculty, University of the Basque Country (UPV/EHU), Donostia-San Sebastian, Spain, 5 Society, Sports and Physical Exercise Research Group (GIKAFIT), Department of Physical Education and Sport, Faculty of Education and Sport, University of the Basque Country (UPV/EHU), Vitoria-Gasteiz, Spain

* olaizola525@gmail.com

**Data Availability Statement:** The data supporting this study is available in the Kaggle public repository (https://www.kaggle.com/datasets/aratzolaizola/vff-update) and can also be obtained upon reasonable request to the corresponding

## Abstract

The aim of the present study it's a providing an update of the profile of very high-speed running (VHSR) of professional female football players. In this study 23 official matches of the Spanish First Division of Women´s Football were analysed. A total of 15 players participated, who were classified according to their location on the matches played: central-corridor (CCP, n = 7), and lateral-corridor (LCP, n = 8). The variables recorded were: 1) Very High-Speed Running (VHSR), considering the accumulated distance, duration, and frequency, and the individual characteristics of the efforts (distance and duration); 2) Recovery between efforts (VHSRe) and; 3) VMAX. The results show that LCP described higher values in accumulative VHSR and VHSR by distance bands, with greater distances, durations and VMAX of VHSRe, with less recovery between efforts. There seems to be an evolutionary trend in the distribution of the efforts, with a higher % of efforts in the 10–20 m band. The results provide useful information that coaching staff of professional women's football teams could use to design specific very high-speed running training to optimize performance in competition.

## Introduction

The rapid growth of women's football in the last two decades has been accompanied, due to scientific interest, by a considerable increase in publications [1–3]. The main topics of study have been strength training and injury incidence [3], although in recent years the detailed description of the conditional demands of the players in the training and competition processes has acquired great prominence, mainly due to the expansion of the use of Global Positioning Systems or GPS [4]. An increasingly deeper knowledge of the demands which female players are subjected to in competition will help configure more appropriate and efficient intervention strategies that will help optimize their performance [5].

author through the repositories of the ELEKIN research group at the University of the Basque Country: elekin.net and ehubox.

**Funding:** The University of the Basque Country, the Basque Government, Engineering and Society and Bioengineering Research Groups, IT1489-22, EUSK22/17, COLAB22/15, PES22/30. "Role of Funder statement: The funders had no role in study design, data collection and analysis, decision to publish, or preparation of the manuscript.

**Competing interests:** No potential conflict of interest was reported by the authors.

The increase in professionalism in women's football is justified, among other things, by the need to compensate for the greater dedication to their preparation, which involves greater exposures of the players to both training loads and participation in a larger number of competitions and/or longer competitions [6]. This improved preparation of the players has resulted in the evolution of physical performance in competition represented, for example, by an increase in the total traveled distance, from approximately 9.0 km during the decade of the 90s [7], to approximately 10.4 km, 20 years later [8], which can reach more than 11.1 km for central midfielders [9]. In this sense, it would be interesting to assess if the increase in the total covered distances is also reflected in the high-speed and sprint speed ranges in competition by the female players, as it has been described in a recent study in men's football [10].

The demand at high-speed seems to be a differential factor in the performance of players in competition, because professional players who compete in domestic leagues travel around 0.5 km at a speed above 19.0 km/h and 125 m at more than 22.5 km/h [11], which is even higher in international competitions [7, 8]. Furthermore, these demands vary depending on the position of the players in the team system, with forward and wide players being the ones who accumulate the largest distances at high-speed and sprint (512–850 m and 154–187 m, respectively) compared to central defenders and midfielders (316–484 m and 59–96 m, respectively) [11, 12].

Trying to provide more detailed information on the profiles of the players with regard to high-speed and sprint running, more than 10 years ago [13] described some interesting aspects of locomotive demand. For example, the average distance for each sprint (>18 km/h) was approximately 15 m, with a duration of 2.3 s and up to 2.5 minutes of lower intensity activity between efforts. Years later, in another study carried out by [12], they described that the players completed a total of 376 high-speed runs (between 12.2 and 19.4 km/h) and 70 sprints (>19.4 km/h) during a match, with a large proportion performed over distances less than 10 m with 14 s between high-speed runs and 1.5 minutes between sprints. In another study carried out the same year [14], it was found out that the players accumulated 17 sprint running's (>20 km/h) with an average distance of 15 m, with a duration of 2.5 s and a recovery between efforts of up to 5.8 minutes. Due to the rapid growth of women's football [6], it might be convenient to update such detailed information on the profile of sprinting in women's professional football.

Therefore, the aim of the present study is to provide an update of the profile of very high-speed running (VHSR, >19 km/h) of professional female football players during official competitive matches. The results of this study could be used as reference values to help to program training sessions and design specific tasks detailing the distance, duration and frequency of the accumulated VHSR and each of the VHSR efforts, adapted to the particular needs of the position of the player.

## Materials and methods

### Study design

An observational study was carried out to describe the profile of running during competition at very high-speed by professional football players competing in the Spanish Women's First Division (LigaF). The records of the 23 matches studied were collected from 17 September 2022 to 20 May 2023, during the 2022–2023 season.

### Participants

A total of 15 players participated in the study (age: 27.1 ±4.9 years; height: 167.0 ±5.2 cm; weight: 58.7 ±4.6 kg; skin folds (i.e., the sum of six skin folds: triceps, subscapularis,

supraspinal, abdominal, anterior thigh and medial calf): 54.5 ±12.8 mm). The players were classified according to their usual location on the field: on the central axis of the field and on the wings or corridors. The central corridor is the area that covers the width of the large area (approximately 40 m), where the central defense (n = 3) and midfielder (offensive and defensive, n = 4) demarcations are included. All these demarcations are defined as central corridor player (CCP). The lateral corridor comprises the areas on the margins of the central corridor, both to the left and to the right. Lateral corridor players (LCP) include the positions of lateral defense (n = 2), lane (n = 2), extreme (n = 2) and forwards with a tendency to play on the wing (n = 2). The players usually did two hours of training a day for four or five days per week, plus a competitive match weekly.

All the players involved in the study signed a written informed consent form. The Ethics Committee of research with humans (CEISH) of the University of the Basque Country (UPV/ EHU) gave its institutional approval of the study (code *M10-2019-099)*.

## Physical variables

The variable Very High-Speed Running (VHSR) is related to the movements of the players that exceeded the threshold of 19 km/h, similar to previous studies [10, 12].

On the one hand, we considered the accumulated values of that variable throughout the match in distance (mVHSR), duration (sVHSR), and frequency (nVHSR). On the other hand, each of the efforts made by the players in that speed range (VHSRe) was described, considering the distance (mVHSRe) and the duration (sVHSRe) of the effort. In addition, the duration of the recovery between efforts (sREC) and the maximum speed (VMAX) reached by the player during the match were recorded. Finally, similar to what was established by [12], VHSR efforts were also described based on different bands of traveled distance: lower than 10 m (0-10m), between 10 and 20 m (10-20m), between 20 and 30 m (20 -30m), between 30 and 40m (30-40m) and larger than 40m (>40m).

## Match analysis

In this study, only the activity profiles of those players who completed the entire match were included, with a total of 123 records (CCP = 80 and LCP = 43), in 23 official competition matches (friendly not included). The playing fields of the matches played at home and away had very similar dimensions (103x65 m). The play style of the team was usually characterized by leaving possession to the rival team based mainly on a style with quick transitions and only occasionally developing a possession game (e.g., with a disadvantage on the scoreboard).

## Procedures

In order to gather position data, the players were monitored with X7 GPS devices (Catapult Sports, Melbourne, Australia) using the Global Positioning System (GPS). The players were familiar with the devices. Each device was placed on the surface between the scapulae, in a pocket carried by an adjustable harness. The devices were activated 10–15 min before the start of the warm-up. To avoid possible differences between devices, during the entire registration period each player used the same device [14, 15].

The download of the records was performed using the software Openfield™ v.3.7.3 (Catapult Sports, Melbourne, Australia). Once the reports were gathered per player and match, the data were imported into a Microsoft Excel spreadsheet (Microsoft Corporation, Washington, USA). Only the players who completed the match were included in the matrix that would be used later to carry out the analyses.

## Statistical analysis

Descriptive statistical data from variables were presented using mean and standard deviation with 95% confidence intervals. Tests for normality (Shapiro-Wilk) and equality of variances (Levene) were applied. The null hypothesis was accepted because the distribution of the data met the normality criterion. Furthermore, the variances were homogeneous. Differences between central corridor and lateral corridor were analyzed by T-test and Bonferroni. The effect size was calculated using Cohen's d (d): A Cohen's d effect size of d = 0.2 was considered a small effect size; an effect size of d = 0.5 was considered a medium effect size; and an effect size of d = 0.8 was considered a large effect size [16]. The level of significance was set at $p < 0.05$. The statistical analysis was performed using the software JASP 0.14.1 (University of Amsterdam, Amsterdam, Kingdom of the Netherlands) and a customized Microsoft Excel spreadsheet (Microsoft Corporation, Washington, USA) for Windows.

## Results

### Accumulated VHSR

LCP accumulated a larger number of VHSR efforts than CCP (43.3 ±11.8 vs. 25.5 ±9.2, mean and standard deviation). In addition, they also accumulated larger total distance and duration of VHSR, with a large magnitude in all the cases (-1.9 to 1.7) (Fig 1).

### Accumulated VHSR by distance bands

LCP reached significantly higher values in nVHSR, mVHSR, and sVHSR compared to CCP in all bands except >40m, with a large magnitude (-1.4 to -1.0) (Table 1).

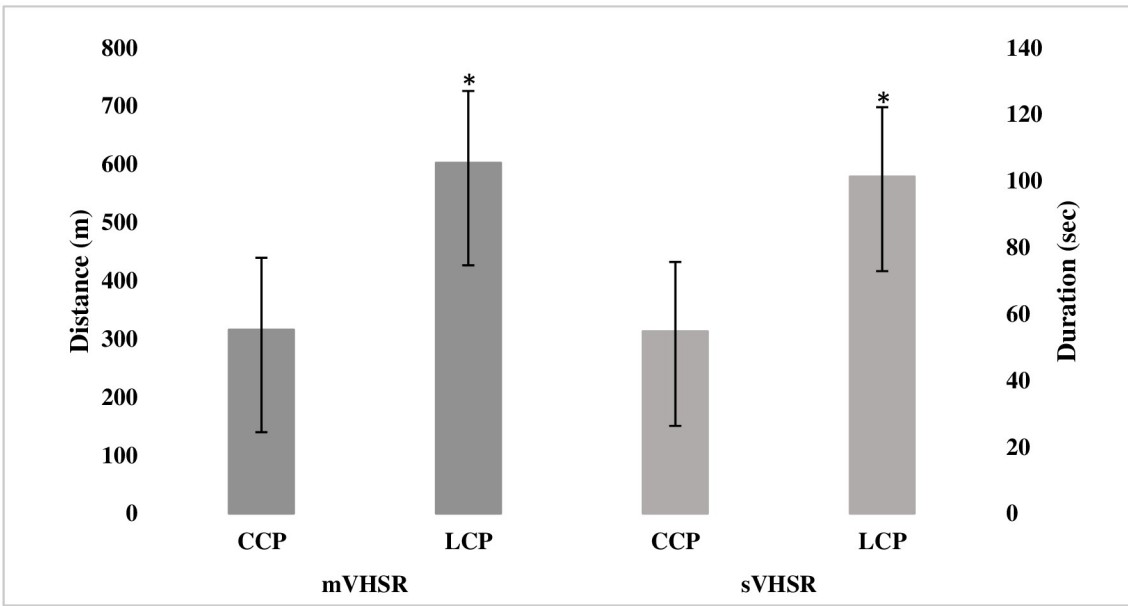

**Fig 1. Accumulated distance and duration of VHSR (mean and standard deviation) in competition according to the playing position.** Note: mVHSR are the total meters covered, and sVHSR is the total duration of the efforts accumulated in the match. CCP is central corridor player and LCP is lateral corridor player. Significant differences (p<0.05) are represented as *.

**Table 1. Means, ± standard deviations, effect sizes (ES) and 95% confidence intervals for number, distance and duration of very high-speed runs at different distances according to playing position.**

| Variables | Position | 0-10m | 10-20m | 20-30m | 30-40m | >40m |
|---|---|---|---|---|---|---|
| nVHSR (n) | CCP | 13.2 ±5.8 | 7.7 ±3.1 | 3.4 ±2.0 | 1.6 ±1.0 | 1.5 ±0.9 |
| | LCP | 20.5 ±6.3* | 12.9 ±4.6* | 5.7 ±2.3* | 3.0 ±1.4* | 1.7 ±1.0 |
| | | ES: -1.2 | ES: -1.4 | ES: -1.1 | ES: -1.1 | |
| | | (-1.6/-0.8) | (-1.8/-1.0) | (-1.5/-0.7) | (-1.6/-0.7) | |
| mVHSR (m) | CCP | 74.0 ±28.5 | 109.4 ±44.6 | 83.3 ±49.2 | 55.7 ±33.0 | 75.4 ±49.0 |
| | LCP | 115.9 ±36.2* | 185.6 ±65.3* | 137.3 ±56.7* | 100.7 ±48.3* | 82.5 ±46.9 |
| | | ES: -1.3 | ES: -1.4 | ES: -1.0 | ES: -1.1 | |
| | | (-1.7/-0.9) | (-1.9/-1.0) | (-1.4/-0.6) | (-1.6/-0.7) | |
| sVHSR (s) | CCP | 13.9 ±5.4 | 19.1 ±5.4 | 13.8 ±8.2 | 9.1 ±5.3 | 12.2 ±8.0 |
| | LCP | 21.6 ±6.7* | 31.9 ±11.1* | 22.3 ±9.1* | 15.9 ±7.7* | 12.7 ±7.0 |
| | | ES: -1.3 | ES: -1.4 | ES: -1.0 | ES: -1.0 | |
| | | (-1.7/-0.9) | (-1.8/-1.0) | (-1.4/-0.6) | (-1.5/-0.6) | |

Note: nVHSR is the number of efforts, mVHSR are the meters covered during the efforts and sVHSR is the duration of the efforts. CCP is central corridor player and LCP is lateral corridor player. Significant differences ($p<0.05$) are represented as *.

## Very high-speed running efforts (VHSRe)

The average and the maximum distances were significantly higher in LCP compared to CCP with a large magnitude (-1.3 to -0.8). Likewise, the average and the maximum duration were also higher in LCP, with a magnitude ranging from large to medium (-1.0 to -0.6). The average and maximum recovery were higher for CCP, with a magnitude ranging from medium to large (-0.6 to 1.2). Finally, the maximum speeds reached were higher for LCP than for CCP with a large magnitude (-1.5) (Table 2).

## Very high-speed running efforts (VHSRe) by distance bands

CCP reached significantly higher values in average duration for the 30-40m band compared to LCP, with a small magnitude (0.4). In addition, CCP also achieved a higher average recovery in the 0-10m, 10-20m and 20-30m bands, with a magnitude ranging from medium to large (-0.6 to 1.2). LCP had higher VMAX values compared to CCP in all bands (Table 3).

**Table 2. Means, ± standard deviations, effect sizes (Cohen´s d) and 95% confidence intervals (95% CI) for the characteristics of the efforts at very high-speed running according to playing position.**

| Variables | Position | | p | Cohen's d | 95% CI for Cohen's d | |
|---|---|---|---|---|---|---|
| | CCP | LCP | | | Lower | Upper |
| Minimum distance (m) | 2.5 ±0.9 | 2.2 ±0.9 | 0.111 | 0.3 | -0.1 | 0.7 |
| Average distance (m) | 12.4 ±2.2 | 13.9 ±1.6* | <0.001 | -0.8 | -1.6 | -0.4 |
| Maximum distance (m) | 35.4 ±10.3 | 47.9 ±8.0* | <0.001 | -1.3 | -1.7 | -0.9 |
| Average duration (s) | 2.2 ±0.4 | 2.3 ±0.2* | 0.002 | -0.6 | -1.0 | -0.2 |
| Maximum duration (s) | 5.8 ±1.6 | 7.4 ±1.2* | <0.001 | -1.0 | -1.4 | -0.6 |
| Average recovery (s) | 314.3 ±127.6* | 179.0 ±58.1 | <0.001 | 1.2 | 0.8 | 1.6 |
| Maximum recovery (s) | 1345.5 ±389.7* | 1155.7 ±143.0 | 0.003 | 0.6 | 0.2 | 1.0 |
| VMAX (Km/h) | 25.7 ±1.6 | 27.9 ±1.3* | <0.001 | -1.5 | -1.9 | -1.1 |

Note: CCP is central corridor player and LCP is lateral corridor player. Significant differences ($p<0.05$) are represented as *.

**Table 3. Means, ± standard deviations, effect sizes (ES) and 95% confidence intervals for the characteristics of the efforts at very high-speed running at different distances according to playing position.**

| Variables | Position | 0-10m | 10-20m | 20-30m | 30-40m | >40m |
|---|---|---|---|---|---|---|
| Average distance (m) | CCP | 5.7 ±0.7 | 14.3 ±1.1 | 24.4 ±2.0 | 33.7 ±2.4 | 47.3 ±5.4 |
| | LCP | 5.7 ±0.6 | 14.5 ±0.8 | 24.2 ±1.3 | 33.9 ±2.0 | 47.2 ±5.2 |
| Average duration (s) | CCP | 1.1 ±0.1 | 2.5 ±0.2 | 4.1 ±0.3 | 5.5 ±0.5* | 7.7 ±0.9 |
| | LCP | 1.1 ±0.1 | 2.5 ±0.1 | 3.9 ±0.2 | 5.3 ±0.3 | 7.3 ±0.8 |
| | | | | | ES:0.4 | |
| | | | | | (0.0/0.9) | |
| Average recovery (s) | CCP | 306.5 ±124.3* | 324.6 ±212.5* | 263.3 ±212.3* | 346.1 ±436.5 | 281.0 ±204.1 |
| | LCP | 170.9 ±71.8 | 177.6 ±88.5 | 162.1 ±107.1 | 210.8 ±180.4 | 234.4 ±250.7 |
| | | ES: 1.2 | ES: 0.8 | ES: 0.6 | | |
| | | (0.8/1.6) | (0.4/1.2) | (0.2/0.9) | | |
| VMAX | CCP | 22 ±1.2 | 24.1 ±1.5 | 25.1 ±1.6 | 24.8 ±1.9 | 24.9 ±1.9 |
| | LCP | 23.1 ±0.9* | 25.4 ±1.0* | 26.6 ±1.6* | 27.1 ±1.9* | 26.9 ±2.0* |
| | | ES: -0.7 | ES: -0.8 | ES: -0.5 | ES: -1.0 | ES: -0.9 |
| | | (-1.1/-0.3) | (-1.1/-0.4) | (-0.9/-0.1) | (-1.4/-0.5) | (-1.5/-0.4) |

Note: CCP is central corridor player and LCP is lateral corridor player. Significant differences (p<0.05) are represented as *.

## Discussion

The aim of the present study is to update the profile of very high-speed running (>19 km/h) of professional female football players during official competitive matches in relation to their position on the field. The main results were: 1) LCP accumulated a larger number, distance and duration of total VHSR and VHSR by distance band, except for the >40m band; 2) LCP showed greater distances, durations and VMAX of the VHSRe with less recovery between efforts; 3) LCP described higher speed peaks in all distance bands, with less recovery between efforts in the 0-10m, 10-20m and 20-30m bands.

The accumulated mVHSR was almost double in the LCP compared to the CCP (603 vs 316 m), and therefore, the accumulated time (sVHSR) was also larger (101 vs 55 s). These results were similar to those reported by [12, 17] where the wing demarcations accumulated between 512 and 850 m, and for the central axis between 316 and 484 m. Regarding the number of accumulated efforts (nVHSR), the LCP also obtained higher values compared to the CCP (43 vs 25, respectively), coinciding with the results obtained in the study of [13, 14], where the central axis players had lower values than the wing players (31–36 vs 43 and 8–21 vs 21–22, respectively). In this sense, the results obtained were similar, with the wing players presenting higher values than the central axis players. However, these findings should be interpreted with caution, because the speed thresholds, recording systems, and sample size were not the same. For instance, the authors did not agree on the lower threshold to consider the high-speed range, being >18 km/h for [13], >19.4 km/h for [12], and >20 km/h for [14]. With regard to the recording systems used to capture the locomotor response at high-speed, they did not coincide either (Optical Player Tracking for [12], and GPS for [13, 14]). Finally, it should be noted that the sample size in relation to records by position and game system varied depending on the study.

Regarding to accumulated VHSR by distance bands, the results of the present study describe that the LCPs reached significantly higher values in nVHSR, mVHSR and sVHSR compared to CCP in all bands except >40m, coinciding with the findings obtained by [12], where the wing players presented higher values than the central axis players in the number of

efforts and in the majority of bands. 50% of the efforts at high-speed were made in the 0-10m band, followed by the 10-20m band that accumulated 30% of the efforts. The accumulation of total meters in the bands was similar (0-10m, 19%, 10-20m, 30%, 20-30m, 22%, 30-40m, 16%, and >40m, 13%), except for the distance of 10-20m that accumulated a higher percentage than the rest. These results were not similar to those found out by [12], because the largest number of efforts was made in the 0-10m band (71–78%). This distribution of efforts may be influenced by contextual variables such as opponent level, match location, or score, which could affect play style [18]. The team was mainly defensive and not very dominant, with a narrow and shallow block, making coverage and short-distance defensive jumps.

Regarding the characteristics of efforts (VHSRe), the LCP made an average distance larger than the CCP, covering 14.0 m in 2.3 s vs. 12.0 m in 2.2 s, respectively. These results were similar to those reported by [13, 14], where the wing players covered distances larger than the central axis players (15.0 m in 2.3 s vs 14.0 m in 2.2 s and 17.0 m in 2.7 s vs 15.0 m in 2.4 s, respectively). Average distance in both studies ranged between 14.0 and 17.0 m, far from the 8.0 m described by [12]. The maximum distances were also higher for the LPCs compared to the CCPs, accumulating 48.0 m and 7.4 s vs 35.0 m and 5.8 s, respectively, with slightly higher values for the LCP compared to those reported by [12], where the traveled distance was 39.0 m.

Due to a lower amount of effort made by the CCP compared to the LCP, the average (5.2 vs 2.9 minutes) and maximum (22.4 vs 19.2 minutes) recovery time between efforts was larger for CCP. These results agree with those reported by [13] (2.5–2.8 vs 2.1 minutes) and [14] (4.7–8.8 vs 4.1–4.4 minutes), where the central players also had longer recovery times. The values registered in the present study differ also from those obtained by [12], with a minimum recovery time of 1.4 minutes and a maximum recovery time of 6.6 minutes. These differences could be due to a greater conditional demand for the Australian league compared to that of the Brazil [13], United States [14], or Spanish league in the present study.

Regarding the maximum speeds reached, wing players obtained higher values than the central axis ones (25.7 km/h vs 27.9 km/h). One of the reasons that could justify these differences between wing players and the central axis players could be the tactical roles that those play in the game system.

It should be noted that this is the first study that describes the characteristics (distance, duration, average recovery time and maximum speeds) of the efforts by distance bands, with no differences found between the LCP and CCP neither in distance or in duration, except in the 30-40m band where the CCP needed a longer time (5.5 vs 5.3 s) to accomplish the same meters (33.7 vs 33.9 m) as the LCP. The CCP accumulated a longer recovery time than the LCP in the first three bands (0-10m, 10-20m and 20-30m), and LCPs presented higher maximum speeds on all bands compared to the CCP, which would support the results obtained in previous studies where the wing players were faster than the players of the central axis [12, 19].

The results of the present study provide useful information that coaching staff of professional women's football teams could use to design specific very high-speed running training to optimize performance in competition. Furthermore, it reinforces the need for coaches to take an individual point of view in the intervention process, associated with playing positions or tactical roles, where variation in competitive demand is significant. In this sense, the results of the present study recommend that central corridor players (CCP) should train to be able to do 25 efforts, of 12 m with 5.2 minutes of recovery, and wing players (LCP) should train to do 50 efforts, 14 m with 2.9 minutes of recovery. In any case, considering the standard deviation of the values, the tasks should allow the possibility of high variability in the locomotor response with respect to distances, frequencies, durations, and recoveries between efforts. Finally, due to the described differences in demands depending on the corridor where they are located, the double accumulated distance in VHSR for wing players (approximately 600 m) compared to

center players (approximately 300 m), the coaching staff should take this into consideration when proposing a change in the player's usual position, either during the match itself or from match to match, to avoid demand peaks in players who are not used to training and playing in the lateral positions of the playing field, being more easily adaptable from wing to center.

However, this study has some limitations. The first one is related to the sample size (one team), so we must be cautious when trying to generalize the results to other teams and categories of women's football. More case studies are necessary to consolidate more solid knowledge regarding the profile of high-speed running in women's professional football. The second limitation relates to the difficulties in comparing the results of the present study with the scarce literature in the field of elite women's football. The chosen samples, the technologies used in the recording, and the thresholds used to delimit the high-speed ranges of other studies may have biased the discussion of the results of the present study. Before continuing to describe physical performance in elite football, it seems necessary to try to agree on the absolute speed thresholds, or propose other alternatives by relative perspectives (e.g., relative speeds based on individual maximum) so that comparisons could be made in a more contextualized way with regard to the physical performance of players.

## Conclusions

The main conclusions of the study were: 1) LCP had higher values than CCP in accumulated VHSR (also by distance band, except for the last one), and in most characteristics of VHSRe, however, it cannot be concluded that there has been an evolution of this speed range in the last years; 2) The characteristics of VHSRe by distance bands were similar for both demarcations except for the average recovery which was higher for CCP in the first three bands (0-10m, 10-20m and 20-30m) and VMAX which was higher for LCP in all bands; 3) The distribution of nVHSR in accumulated VHSR by distance bands was similar for both demarcations, and it may be an evolutionary tendency to accumulate larger number of efforts in the second band compared to those reported in the literature (i.e., 10-20m).

## Acknowledgments

The authors would like to thank all the football players who participated in this study, IT1489-22, PID2023-147577NB-100, EUSK22/17, COLAB22/15, PES22/30.

## Author Contributions

**Conceptualization:** Aratz Olaizola, Ibai Errekagorri, Julen Castellano.

**Data curation:** Aratz Olaizola, Karmele Lopez-de-Ipina, Julen Castellano.

**Formal analysis:** Aratz Olaizola, Ibai Errekagorri, Karmele Lopez-de-Ipina, Julen Castellano.

**Funding acquisition:** Aratz Olaizola.

**Investigation:** Aratz Olaizola, Ibai Errekagorri, Karmele Lopez-de-Ipina, Pilar María Calvo, Julen Castellano.

**Methodology:** Aratz Olaizola, Ibai Errekagorri, Karmele Lopez-de-Ipina, Julen Castellano.

**Project administration:** Aratz Olaizola.

**Resources:** Aratz Olaizola.

**Software:** Aratz Olaizola.

**Supervision:** Aratz Olaizola, Ibai Errekagorri, Julen Castellano.

**Validation:** Aratz Olaizola, Ibai Errekagorri, Karmele Lopez-de-Ipina, Julen Castellano.

**Visualization:** Aratz Olaizola, Karmele Lopez-de-Ipina, Julen Castellano.

**Writing – original draft:** Aratz Olaizola, Ibai Errekagorri, Karmele Lopez-de-Ipina, Pilar María Calvo, Julen Castellano.

**Writing – review & editing:** Aratz Olaizola, Karmele Lopez-de-Ipina, Pilar María Calvo, Julen Castellano.

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
