## [Decision Letter · Decision Letter 0]

9 May 2024

PONE-D-24-06432Very High-Speed Running (VHSR) profile in elite female football: an updatePLOS ONE

Dear Dr. Olaizola,

Thank you for submitting your manuscript to PLOS ONE. After careful consideration, we feel that it has merit but does not fully meet PLOS ONE’s publication criteria as it currently stands. Therefore, we invite you to submit a revised version of the manuscript that addresses the points raised during the review process.  I agree with Reviewer 2, and you should make a minor revision of your manuscript, regarding his comments.

We look forward to receiving your revised manuscript.

Kind regards,

Jovan Gardasevic

Academic Editor

PLOS ONE

 [The University of the Basque Country, the Basque Government, Engineering and Society and Bioengineering Research Groups, IT1489-22, EUSK22/17, COLAB22/15, PES22/30.].  

Additional Editor Comments:

Dear author,

I agree with Reviewer 2, and you should make a minor revision of your manuscript, regarding his comments.

Kind,

reviewer

Reviewers' comments:

Reviewer's Responses to Questions

**Comments to the Author**

1. Is the manuscript technically sound, and do the data support the conclusions?

Reviewer #1: Yes

Reviewer #2: Yes

2. Has the statistical analysis been performed appropriately and rigorously? 

Reviewer #1: Yes

Reviewer #2: Yes

3. Have the authors made all data underlying the findings in their manuscript fully available?

Reviewer #1: Yes

Reviewer #2: Yes

4. Is the manuscript presented in an intelligible fashion and written in standard English?

Reviewer #1: Yes

Reviewer #2: Yes

5. Review Comments to the Author

Reviewer #1: very interesting study. I must emphasize that the order is methodologically very good. the results are clearly presented, the discussion is very detailed and interesting.

I recommend a paper prepared in this way for publication.

Reviewer #2: This topic has excellent, valuable insights into elite female football players' very high-speed running profiles on both the theoretical and practical sides. Therefore, you selected a critical topic that is especially significant for the coaching staff of professional women's football teams: crucial for optimizing training and performance strategies in professional women's football. You mentioned that this research has seen a considerable increase in publications. Did you explore all relevant references? The research background should be presented in more detail, adding depth to the findings by considering contextual factors.

6. PLOS authors have the option to publish the peer review history of their article (what does this mean?). If published, this will include your full peer review and any attached files.

Reviewer #1: No

Reviewer #2: **Yes: **Radenko M. Matic

---

## [Author Response · Author response to Decision Letter 0]

5 Jul 2024

RESPONSES TO REVIEWER 1

Very interesting study. I must emphasize that the order is methodologically very good. the results are clearly presented, the discussion is very detailed and interesting.

Reviewer #1: I recommend a paper prepared in this way for publication.

We thank the Reviewer for his/her valuable comments and suggestions. 

RESPONSES TO REVIEWER 2

We thank the Reviewer for his/her valuable comments and suggestions. We have corrected and introduced all your comments, corrections and suggestions and we answer below to your questions. 

Next, we provide our responses to each of the comments with a detailed description of how our manuscript was modified accordingly.

Reviewer #2: This topic has excellent, valuable insights into elite female football players' very high-speed running profiles on both the theoretical and practical sides. Therefore, you selected a critical topic that is especially significant for the coaching staff of professional women's football teams: crucial for optimizing training and performance strategies in professional women's football. 

You mentioned that this research has seen a considerable increase in publications. Did you explore all relevant references? The research background should be presented in more detail, adding depth to the findings by considering contextual factors.

Thank you for your comments and suggestions, we agree. We corrected it in the manuscript.

This distribution of efforts may be influenced by contextual variables such as opponent level, match location, or score, which could affect play style [19]. The team was mainly defensive and not very dominant, with a narrow and shallow block, making coverage and short-distance defensive jumps.

Castellano J, Blanco-Villaseñor A, Álvarez D. Contextual Variables and Time-Motion Analysis in Soccer. Int J Sports Med. 2011;32: 415–421. doi:10.1055/s-0031-1271771

Díaz-Serradilla E, Castillo D, Rodríguez-Marroyo JA, Raya González J, Villa Vicente JG, Rodríguez-Fernández A. Effect of Different Nonstarter Compensatory Strategies on Training Load in Female Soccer Players: A Pilot Study. Sports Health: A Multidisciplinary Approach. 2023;15: 835–841. doi:10.1177/19417381231176555

---

## [Editor Report · Decision Letter 1]

29 Jul 2024

Very High-Speed Running (VHSR) profile in elite female football: an update

PONE-D-24-06432R1

Dear Mr Aratz Olaizola,

We’re pleased to inform you that your manuscript has been judged scientifically suitable for publication and will be formally accepted for publication once it meets all outstanding technical requirements.

Kind regards,

Jovan Gardasevic

Academic Editor

PLOS ONE

Additional Editor Comments (optional):

Dear Author,

You revised the manuscript against the comments of reviewer 2, and I think that the manuscript is now ready for publication in this respected journal.

Kind regards,

Academic Editor
---

## [Editor Report · Acceptance letter]

7 Aug 2024

PONE-D-24-06432R1 

PLOS ONE

Dear Dr. Olaizola, 

I'm pleased to inform you that your manuscript has been deemed suitable for publication in PLOS ONE. Congratulations! Your manuscript is now being handed over to our production team.

Kind regards, 

on behalf of

Dr. Jovan Gardasevic 

Academic Editor

PLOS ONE